


# Atmospheric $CH_4$ and $CO_2$ enhancements and biomass burning emission ratios derived from satellite observations of the 2015 Indonesian fire plumes

Robert J. Parker[1,3], Hartmut Boesch[1,3], Martin J. Wooster[2,3], David P. Moore[1,3], Alex J. Webb[1], David Gaveau[4], and Daniel Murdiyarso[4,5]

[1]Earth Observation Science, Department of Physics and Astronomy, University of Leicester, Leicester, UK
[2]King's College London, Department of Geography, London, UK
[3]NERC National Centre for Earth Observation, UK
[4]Center for International Forestry Research, P.O. Box 0113 BOCBD, Bogor 16000, Indonesia.
[5]Department of Geophysics and Meteorology, Bogor Agricultural University, Bogor 16680, Indonesia

*Correspondence to:* Robert Parker (rjp23@le.ac.uk)

**Abstract.**

The 2015-2016 strong El Niño event has had a dramatic impact on the amount of Indonesian biomass burning, with the El Niño driven drought further desiccating the already drier than normal landscapes that are the result of decades of peatland draining, widespread deforestation, anthropogenically-driven forest degradation, and previous large fire events. It is expected

that the 2015-16 Indonesian fires will have emitted globally significant quantities of greenhouse gases (GHGs) to the atmosphere, as did previous El Niño driven fires in the region. The form which the carbon released from the combustion of the vegetation and peat soils takes has a strong bearing on its atmospheric chemistry and climatological impacts. Typically, burning in tropical forests and especially in peatlands is expected to involve a much higher proportion of smouldering combustion than the more flaming-characterised fires that occur in fine-fuel dominated environments such as grasslands, consequently

producing significantly more $CH_4$ (and CO) per unit of fuel burned. However, currently there have been no aircraft campaigns sampling Indonesian fire plumes, and very few ground-based field campaigns (none during El Niño), so our understanding of the large-scale chemical composition of these extremely significant fire plumes is surprisingly poor compared to, for example, those of southern Africa or the Amazon.

Here, for the first time, we use use satellite observations of $CH_4$ and $CO_2$ from the Greenhouse gases Observing SATel-

lite (GOSAT) made in large scale plumes from the 2015 El Niño-driven Indonesian fires to probe aspects of their chemical composition. We demonstrate significant modifications in the concentration of these species in the regional atmosphere around Indonesia, due to the fire emissions.

Using CO and fire radiative power (FRP) data from the Copernicus Atmosphere Service, we identify fire-affected GOSAT soundings and show that peaks in fire activity are followed by subsequent large increases in regional greenhouse gas concentra-

tions. $CH_4$ is particularly enhanced, due to the dominance of smouldering combustion in peatland fires, with $CH_4$ total column values typically exceeding 35 ppb above that of background "clean air" soundings. By examining the $CH_4$ and $CO_2$ excess concentrations in the fire-affected GOSAT observations, we determine the $CH_4/CO_2$ fire emission ratio for the entire 2-month





period of the most extreme burning (September- October 2015), and also for individual shorter periods where the fire activity temporarily peaks. We demonstrate that the overall $CH_4$ to $CO_2$ emission ratio (ER) for fires occurring in Indonesia over this time is 6.2 ppb/ppm. This is higher than that found over both the Amazon (5.1 ppb/ppm) and southern Africa (4.4 ppb/ppm), consistent with the Indonesian fires being characterised by an increased amount of smouldering combustion due to the large

amount of organic soil (peat) burning involved. We find the range of our satellite-derived Indonesian ERs (6.18 ppb/ppm to 13.6 ppb/ppm) to be relatively closely matched to that of a series of "close-to-source" ground-based sampling measurements made on Kalimantan at the height of the fire event (7.53 to 19.67 ppb/ppm), although typically the satellite-derived quantities are slightly lower on average. This seems likely to be because our field sampling mostly intersected smaller-scale peat burning plumes, whereas the large-scale plumes intersected by the GOSAT TANSO-FTS footprints would very likely come from burn-

ing that was occurring in a mixture of fuels that included peat, tropical forest and already cleared areas of forest characterised by vegetation types that are more fire-prone than the natural rainforest biome (e.g. post-fire areas of ferns and scrubland, along with agricultural vegetation)

The ability to determine large-scale emission ratios from satellite data allows the combustion behaviour of very large regions of burning to be characterised and understood in a way not possible with ground-based studies, and which can be logistically

difficult and very costly to consider using aircraft observations. We therefore believe the method demonstrated here provides a further important tool for characterising biomass burning emissions, and that the GHG emission ratios derived for the first time for these large-scale Indonesian fire plumes during an El Niño event, points the way to more routinely assessing spatio-temporal variations in biomass burning emission ratios using future satellite missions that will have more complete spatial sampling than GOSAT, and that will enable the contributions of these fires to the regional atmospheric chemistry and climate

to be better understood.

# 1   Introduction

The 2015-2016 strong El Niño event, which is ongoing in the tropical Pacific at the time of writing, has had a dramatic impact on the amount of landscape burning occurring across large parts of Indonesia. Landscape fires are commonly used in this environment to clear forest and help manage land for agriculture, but an El Niño-driven drought has further dried out the

already human-modified landscapes of Central Kalimantan and south Sumatra. These regions are already more flammable than their natural state due to decades of peatland draining and deforestation, anthropogenically-driven forest degradation, as well as the legacy of previous large fire events (Wooster et al., 2012). Even short localised fire events in these environments can lead to significant greenhouse gas (GHG) emissions, as demonstrated by Gaveau et al. (2014) who report that a one week fire event in Riau Province (Sumatra) was responsible for emitting $172 \pm 59$ Tg $CO_2$-eq., approximately 5-10% of Indonesia's average

annual GHG emissions. The 2015 El Niño-driven fire season in Indonesia is already known to have been far more extensive than in 'normal' years (Voiland, 2016), and during the last very strong El nino (1997-98; the strongest yet on record) massive increases in Indonesian fire activity were similarly recorded (Wooster et al., 1998; Page et al., 2002). Indeed all previous El Niño events, back to the next strongest event after 1997-98 (i.e. that of 1982-83), appear to have produced significant increases



in burning over Indonesia, as detailed in Wooster et al. (2012). Whilst the degree of fire activity increase associated with El Niño is possible to gauge using, for example, satellite-derived active fire counts, forest cover change or burned area maps (e.g. Trigg et al. (2006); Langner et al. (2007); Langner and Siegert (2009); Wooster et al. (2012)), what is equally valuable is information on the emissions to the atmosphere resulting from these burns, so that their atmospheric impacts can be more fully

determined.

Using satellite-derived estimates of burned area along with assumptions on peatland 'depths of burn', Page et al. (2002) estimated that the 1997 El Niño-driven Indonesian fires released an amount of carbon (0.81-2.57 Pg) equivalent to between 13 and 40% of that year's annual global carbon emissions from fossil fuels, contributing to the largest annual increase in atmospheric $CO_2$ concentration detected since records began in the 1950s (Wang et al., 2013). More recently, van der Werf

et al. (2010) have reported similarly anomalous estimates for that year's Indonesian burning, based on related methodologies but different datasets, and anomalies in both inter-annual variability and the atmospheric growth rates of $CO_2$ and $CH_4$ continue to be attributed to biomass burning events, including El Niño-driven Indonesian fires (Kasischke and Bruhwiler, 2002; van der Werf et al., 2004; Simmonds et al., 2005). It is possible that the 2015-16 El Niño-driven Indonesian fires, which at the time of writing have largely ceased due to heavy rains (but which may well return in 2016), may ultimately be of a similarly anomalous

magnitude to those driven by prior El Niño events. There therefore exists a strong interest in both quantifying the amount of fire activity occurring and in calculating the overall carbon emissions to the atmosphere that result. Furthermore, the types of biomes being affected are important, because whilst post-fire vegetation regrowth in fire-affected areas does subsequently take up some of the released carbon, areas of burned tropical forest are often replaced by plants holding far less carbon per unit area, and the burning of peat represents an effectively permanent transfer of carbon from the land to the atmosphere

(Page et al., 2002). The form in which the carbon is emitted into the atmosphere also has a strong bearing on the emission impacts, with most carbon being released as either the long-lived GHG carbon dioxide, the shorter-lived but much stronger GHG methane, or the air pollutant carbon monoxide (Andreae and Merlet, 2001). Typically, burning in tropical forests and especially in peatlands is expected to involve a much higher proportion of smouldering combustion than the more flaming-characterised combustion that occurs in fine-fuel dominated environments such as grasslands (Christian et al., 2007; Liu et al.,

2014). Hence, fires in peatlands and tropical forests are expected to produce more CO and $CH_4$ per unit of fuel burned, with a consequent reduction in the amount of $CO_2$. However, currently the only information on emissions make-up in fires in Indonesian biomes come from relatively few lab-based studies where samples of fuels have been burned in combustion chambers (i.e. Christian et al. (2003); Othman and Latif (2013); Liu et al. (2014)). At present there are no known field-based studies of emissions make-up, certainly none conducted during El Niño years where the dry conditions may promote different

combustion behaviour than occurs under more 'normal' meteorological and fuel moisture conditions, and none where the constituents of the large-scale plumes that most likely contain the bulk of the emitted gases (and aerosols) are assessed. The latter point is important because whilst ground-based sampling can measure emission make-up close to source, including in the field under real landscape combustion conditions, such an approach is by necessity limited to capturing smoke from individual fire locations, and usually from smaller fires, and these measurements may not fully represent the emissions characteristics of

the type of large-scale plumes that may actually be responsible for holding most of the combustion products. Aircraft sampling





can provide a means to capture the latter's characteristics (e.g. Yokelson et al. (1999)), but such campaigns are costly, infrequent and logisitically challenging. An alternative approach to characterising the emission make-up of large-scale fire plumes is via satellite based sounding of wildfire plume chemistry, which has so-far been demonstrated only a few times, by Coheur et al. (2009) using IASI onboard METOP and by Ross et al. (2013) using the TANSO-FTS instrument onboard the Greenhouse

gases Observing SATellite (GOSAT). Here we build on the latter work to exploit GOSAT's observations of $CH_4$ and $CO_2$ over the 2015 El Niño-driven Indonesian fires, using these to demonstrate the increase in atmospheric concentrations of $CH_4$ and $CO_2$ associated with the large scale biomass burning plumes, and deriving from these observations the $CH_4$ to $CO_2$ emission ratios for these El Niño driven Indonesian fires for the first time. We compare these Indonesian fire emissions ratios to those derived from GOSAT in alternative tropical biomes having different combustion characteristics (southern African savannah and

the Amazon basin). In combination with Fire Radiative Power (FRP) and atmospheric carbon monoxide data taken from the new Copernicus Atmosphere Service Global Fire Assimilation System (CAMS-GFAS (2016)) we demonstrate the Indonesian fires are occurring in peatland dominated landscapes that explain certain characteristics of the noted emission ratios, which are themselves important in determining the so-called emissions factors representative of the combustion processes occurring in these very large-scale landscape fires.

Emissions factors (EFs) are necessary when converting estimates of the amount of fuel burned (obtained from burned area or FRP-based methods) into a quantity of each trace gas (Koppmann et al., 2005; Reid et al., 2005). These EFs are themselves often calculated through the use of emission ratios (ER) which are determined from the ratio of the excess concentrations emitted from wildfires (Andreae and Merlet, 2001). Whilst the emission factors are only one aspect of calculating the overall emitted amounts, due to the fact that satellite observations of burned area and FRP have significantly improved in recent years,

the accuracy of the emission factors is becoming more crucial to the overall accuracy of the emissions (Van Leeuwen and Van Der Werf, 2011). The capability to measure the $CH_4/CO_2$ ERs from a variety of wildfires in different biomes across the globe, consistently using a single instrument/approach and from very large scale plumes that represent some of the largest individual fires emissions sources, is therefore a significant advancement. We first demonstrated this capability in Ross et al. (2013), and here we focus on extending this determination of satellite-derived $CH_4/CO_2$ ERs to Indonesia during the anomalously large

El Niño-driven fire season of September-October 2015. The emissions ratios themselves are of clear interest in helping to determine the relative amounts of these two key GHGs released by the fires, but also the relative amounts of $CH_4$ and $CO_2$ being released in a smoke plume are known to vary with the dominance of smouldering and flaming combustion of the causal fire, as do the more commonly used $CO_2$ and CO measures (e.g. Lobert and Warnatz (1993), Yokelson et al. (1996), Wooster et al. (2011)). Furthermore, knowledge of the relative amounts of these two phases of combustion are known to exert strong

controls on the relative emissions of many other compounds (e.g. Yokelson et al. (1996), Cofer et al. (1998), Lee et al. (2010)), and thus if we can better understand the relative $CO_2$ and $CH_4$ emissions makeup of the large-scale plumes emanating from these fires this may provide useful information to better estimate the type of combustion occurring and thus potentially the overall emissions characteristics beyond the two species observed.





## 2 El Niño and Indonesian Fire Activity

El Niño describes a large-scale climate anomaly that typically occurs once or twice per decade, with one of the key characteristics being significantly warmer than normal sea surface temperatures (SSTs) in the equatorial eastern Pacific Ocean (Trenberth, 1997). The many other effects associated with an El Niño event are complex and not always consistent between different El Niños, but most events are accompanied by warmer temperatures across much of South America, Africa, South-East Asia and Western Europe, decreased precipitation over central/southern Africa, central America, South-East Asia, and increased precipitation over the southern United States and Western Europe (Hartmann et al., 2013). Indonesia is located in the equatorial region and can be particularly affected by El Niño events, for example usually experiencing warmer temperatures and significant reductions in rainfall that exacerbate certain aspects of a landscape already heavily modified by human actions. In particular, much of the low lying land on the Indonesian islands of Sumatra and Kalimantan that was originally covered by moist, forested peatlands has been cleared and drained for agriculture, and this has led to much drier landscape conditions. Fire is commonly used to manage the land, and during the droughts associated with El Niño the already heavily disturbed peatland landscapes can become so dry that they can be ignited from the vegetation fires that are widespread even in "normal" years (Gaveau et al., 2014). Such fires can burn down into the carbon rich peat for weeks, whilst also spreading across the landscape to ignite new areas - including spreading into areas of remaining tropical forest that normally are not prone to fire. During El Niño these peat and forest fires can thus affect areas that are very much larger than those burned during "normal" years, particularly during the strongest El Niño events when fire activity can be more than an order of magnitude higher (van der Werf et al., 2008; Wooster et al., 2012). As described in Section 1, during the 1997-98 El Niño, fires in Indonesia are estimated to have released huge amounts of carbon in to the atmosphere and because of the smouldering nature of peat (and to some extent tropical forest as well), a greater proportion of these emissions are likely to be in the form of non-$CO_2$ gases, primarily the air pollutant CO and the strong greenhouse gas $CH_4$, than is the case for flaming fires (Christian et al., 2003, 2007). This contrasts with the burning of the El Niño dried finer fuels which will typically burn primarily via flaming combustion and thus release a lower proportion of CO and $CH_4$ and a higher proportion of $CO_2$ whose global warming potential is significantly lower than that of methane (Myhre et al., 2013).

### 2.1 Magnitude of El Niño Events and the Associated Fire Activity

There are many different ways to quantify the magnitude of an El Niño event but one of the most widely accepted is the Multi-Variate ENSO Index (MEI) (Wolter and Timlin, 1998). This is based on observations of a variety of meteorological parameters over the tropical Pacific Ocean. By this metric (Wolter, 2016), the current El Niño event that we are experiencing (2015/2016) is already the third strongest event on record (behind 1997/98 and 1982/83), with the potential to be classified even higher before it is complete.

To investigate the magnitude of the increased fire activity over Indonesia that has been associated with the current El Niño we examined the fire radiative power (FRP) being released from the identified combustion zones. FRP is a measure of a fires release rate of thermal radiation, and is strongly related to the rate of fuel consumption and trace gas and aerosol emission





(Wooster et al., 2005; Freeborn et al., 2008). FRP is therefore both an indicator for the presence of fire, and an estimator for the amount of material being emitted to the atmosphere from that fire. Global satellite observations of FRP are made from the MODIS instruments onboard the NASA Terra and Aqua satellites at a nadir spatial resolution of 1 km, and these are incorporated into the Copernicus Atmosphere Monitoring Services (CAMS) Global Fire Assimilation System (GFAS), set up under the Monitoring Atmospheric Composition and Climate (MACC) series of research projects (CAMS-GFAS (2016)). Using FRP data converted to "FRP density" by dividing by the grid cell area ($0.1°$x$0.1°$) and adjusting for the impact of unseen parts of the land surface due to gaps in satellite coverage and variations in cloud cover (Kaiser et al., 2012), GFAS produces estimates of trace gas emissions from the mapped fire affected areas, which CAMS then uses in its atmospheric chemistry transport model to identify atmospheric abundances of the released chemical species.

Figure 1 shows the monthly total FRP density (in W/m$^2$) over the Indonesian region (defined as 5°N-10°S, 90°E-150°E) for the last seven years calculated from the GFAS data, including adjustments for observation frequency and cloud cover (Kaiser et al., 2012). Whilst significant landscape burning takes place every year between July and October in this Indonesian region, the fires that took place in the latter part of 2015 (particularly September and October 2015) were clearly of an extreme magnitude, with the cumulative FRP density for October 2015 exceeding 7500 W/m$^2$, compared to the second highest value of just over 2000 W/m$^2$ (October 2014).

Whilst FRP gives an indication of the intensity of fires and their associated emissions to the atmosphere, the number of fires is also a useful indicator of fire activity, especially in regions which may see many small fires as opposed to fewer, but larger, events (Wooster and Zhang, 2004; Schroeder et al., 2014). For this reason, the original MODIS MOD14/MYD14 fire counts were also examined (Giglio et al., 2003). The number of fires observed by MODIS across Indonesia during September/October 2015 are shown in Fig. 2. Overlain onto this in green are the locations of known peatlands in Sumatra, Kalimantan and Papua. It is clear that the majority of the most fire affected regions of Indonesia during the September and October 'extreme fire event', i.e. Central Kalimantan and the south-east region of Sumatra, are located in areas dominated by peatlands.

## 2.2 Fire Emissions and Combustion Regimes

As already stated, in contrast to the flaming combustion involved in the burning of wood/grass, peatland fires are typically dominated by deeper smouldering combustion. As smouldering combustion is less efficient than flaming combustion, there is a higher proportion of CO, $CH_4$ and other non-methane hydrocarbons (NHMCs) released compared to $CO_2$ (Bertschi et al., 2003; Yokelson et al., 2008; Wooster et al., 2011).

A literature review of previous ground and aircraft based measurements of the $CH_4/CO_2$ ER indicates a wide range of values, demonstrating the variability that can be dependent on not only the fuel type but also on additional factors such as fuel moisture content, the ratio of living to dead matter and how recently the area last burned (Korontzi et al., 2003). To take just one example, Koppmann et al. (1997) present $CH_4/CO_2$ ER values for flaming fires of 2.6 ppb/ppm from sugar cane fields, increasing to 10.3 ppb/ppm over fires dominated by smouldering combustion in forest and shrubland. Fires with intermediate values were reported to represent a mixture of smouldering and flaming combustion. Similarly, Hurst et al. (1994) report mean ER values of $2.1 \pm 1.5$ ppb/ppm for flaming combustion, $5.3 \pm 2.0$ ppb/ppm for mixed combustion and $10.1 \pm 3.9$ ppb/ppm for



smouldering combustion. A further study, (Bonsang et al., 1995), present values of 3.2-4.6 ppb/ppm for flaming combustion, increasing to 7.8 ppb/ppm for smouldering combustion in savannah/forest regions. The wide range of $CH_4$ to $CO_2$ emission ratios reported by these different studies demonstrates that, even when measured close-to-source as all these were, there is a high degree of variability intrinsic to the $CH_4/CO_2$ ER but the relative behaviour remains consistent, namely that flaming

processes produce smoke with a lower $CH_4$ to $CO_2$ ratio than smouldering processes, and thus it maybe possible to distingush between these two types of combustion using measurments of this ratio.

  The objectives of this work are to first determine whether the expected high concentrations of $CH_4$ emitted by the extreme peatland burning in September/October 2015 over Indonesia are observable from satellite data and if that is the case, to then determine the $CH_4/CO_2$ emission ratio of the resulting large-scale smoke plumes and compare this to measurements made

in-situ. The capability to examine the large-scale emission ratios of a region such as this is important because if GOSAT can measure $CH_4/CO_2$ emission ratios, such observations contain information related to the mix of combustion types occurring and can thus help discriminate predominantly smouldering from predominantly flaming regions. Not only is this of direct interest for the $CH_4$ and $CO_2$ emissions themselves but is also useful when considering the many other species contained within the smoke, because the relative abundance of most of these is in part dependent on the amount of flaming and smouldering

combustion occurring.

  This work is presented as follows. Section 3 introduces the GOSAT satellite data used in this work, providing details on the retrieval method and how the $CH_4/CO_2$ data has been used to determine fire emission ratios. Section 4 describes the methodology used for determining whether a GOSAT sounding is affected by fire and provides statistics on the number of fire-affected soundings that we observe over the Indonesian fire region. Section 5 goes on to examine the enhancement in

$CH_4$ as observed from the fire-affected data while Section 6 then uses these data to determine $CH_4/CO_2$ fire emission ratios, comparing them to in-situ observations of the same El Niño driven fire event. Finally we summarise our findings and comment on the outlook for further study in this area of research.

## 3 GOSAT Proxy $XCH_4$ Data

GOSAT was the first dedicated greenhouse gas measurement mission based on an Earth Observation satellite approach, and

was launched by the Japanese Space Agency (JAXA) on 23rd January 2009 (Kuze et al., 2009). GOSAT is equipped with two instruments. The first is the Thermal And Near infrared Sensor for carbon Observations - Fourier Transform Spectrometer (TANSO-FTS), which provides point-based measurements of total column $CO_2$ and $CH_4$ with near-surface sensitivity because of its use of shortwave infrared (SWIR), as well as a thermal infrared (TIR) band sensitive to the mid-troposphere. The second is the Cloud and Aerosol Imager (TANSO-CAI), which provides multispectral imagery that gives additional cloud/aerosol

information about the region of interest within which the TANSO-FTS measurement footprints fall.

  The TANSO-FTS measurement pattern originally consisted of 5 (later changed to 3) across-track points with a footprint of ∼10.5 km, each separated by approximately 100 km on the ground. GOSAT also has capabilities for agile-pointing, allowing both target mode and observations of the glint spot over the ocean. Near-surface sensitivity to the target gases is achieved by





the TANSO-FTS instrument utilising three SWIR spectral bands at 0.76, 1.6 and 2.0 $\mu$m, with mid-tropospheric sensitivity available from a fourth band operating between 5.5 and 14.3 $\mu$m in the TIR. Kuze et al. (2016) provide extensive details of the performance and operation of the TANSO-FTS instrument over the past 6 years. In short, although GOSAT has experienced three major anomalies over its lifetime (a solar paddle failure in May 2014, a pointing system issue in January 2015, and a cryo-

cooler restart in August 2015) it continues to operate well, providing high-quality atmospheric radiance measurements from which we are able to retrieve dry-air column-averaged fractions of $CO_2$ and $CH_4$ (denoted as $XCO_2$ and $XCH_4$, respectively).

Details of the University of Leicester Proxy $XCH_4$ GOSAT retrieval, including recent updates and uncertainty characterisation, can be found in Parker et al. (2011, 2015). In brief, the retrieval utilises the original Orbiting Carbon Observatory (OCO) "full physics" (so-called as the radiative transfer attempts to explicitly model the physical behaviour of the aerosol-scattered

light) retrieval algorithm (Boesch et al., 2011; Cogan et al., 2012; O'Dell et al., 2012) developed to obtain $XCO_2$ from a simultaneous fit of NIR/SWIR $O_2$ and $CO_2$ bands and subsequently modified to operate on GOSAT spectral data to retrieve $XCH_4$ using the light-path proxy approach. Developed by Frankenberg et al. (2006) for use on SCIAMACHY data, this proxy method utilises the fact that the majority of the influence of atmospheric scattering on the retrieved $XCH_4$ can be negated through the co-retrieval of the spectrally close 1.6 $\mu$m $CO_2$ band, since the signal related to both species undergo the same light-path

enhancement through scattering. The resulting $XCH_4/XCO_2$ ratio is therefore robust to the effects of aerosol. Generally the final Proxy $XCH_4$ is obtained via the application of $XCO_2$ model fields to this ratio. Typically due to the fact that there is significantly less influence from aerosol on the final product than with the typical "full physics" retrieval approach (Butz et al., 2010), high-quality retrievals are possible even under cloud/aerosol conditions where the typical full physics retrieval struggles. Not only does this result in many more successful soundings globally, it also allows studies over cloudy or smoke-affected

regions where no data at all may be available from the typical "full physics" retrieval approach.

$XCH_4$ data obtained using the Proxy approach described above have been used in many inversion studies (Fraser et al., 2013; Wecht et al., 2014; Fraser et al., 2014; Cressot et al., 2014; Alexe et al., 2015; Turner et al., 2015) to estimate both global and regional emissions of $XCH_4$. Normally the main disadvantage of the Proxy $XCH_4$ retrieval is that it requires an accurate and unbiased $XCO_2$ model to convert the ratio back into $XCH_4$ (Schepers et al., 2012). However, in our current study

of the atmospheric impacts and emission ratios of the El Niño driven fires in Indonesia, we use only the individual retrieved $XCH_4$ and $XCO_2$ components of the Proxy retrieval and hence, have no dependence on any $CO_2$ model. For the purposes of this study, the standard GOSAT Proxy data record (typically generated as part of the ESA GHG-CCI project (Buchwitz et al., 2015), 4-6 months behind real time due to the use of ECMWF ERA-Interim data in the processing chain) has been extended with the use of ECMWF Analysis data in order to produce results more quickly than are possible with the normal route. In this

way, the Proxy $XCH_4$ timeseries has been extended from June 2015 to November 2015.

In Section 1, it was shown that September/October 2015 exhibited significantly higher FRP over Indonesia than previous years. Before exploring the GOSAT data over Indonesia in more detail, it is first useful to put the GHG observations for September and October 2015 into the context of the longer GOSAT timeseries. Figure 3 shows the 95th-percentile values for the monthly GOSAT data over Indonesia for the entire data record from April 2009 to November 2015. The upper panel shows

the $XCH_4/XCO_2$ ratio, with the central and lower panels showing the individual $XCH_4$ and $XCO_2$ respectively.



In order to quantify the extreme nature of the October 2015 observations and to account for the annual growth rate, we define the magnitude of the enhancement as the October-July difference for each year. For $CO_2$ we observe a magnitude of 4.35 ppm for October 2015 compared to a mean value of $1.05 \pm 1.42$ ppm for the previous years (2009-2014). In the case of $XCH_4$, the enhancement value for October 2015 is found to be 45.65 ppb compared to an average for previous years of $11.93 \pm 3.60$ ppb. The enhancement of both the $XCO_2$ and, in particular, $XCH_4$ in October 2015 is therefore significantly higher than that observed over the region in previous years, corresponding to the extreme in fire activity observed in Fig. 1.

## 4  Identifying Fire-Affected GOSAT Soundings

Section 3 established that a significant increase is observed in the monthly maximum values for the $XCH_4$, $XCO_2$ and the $XCH_4/XCO_2$ during September/October 2015 (calculated as the 95th-percentile values) recorded over Indonesia by GOSAT. To further investigate the atmospheric GHG anomalies identified over Indonesia by GOSAT in Figure 3, it is first necessary to identify which GOSAT soundings are directly affected by fire emissions, and which can be considered "background" (clear) cases

We use the CAMS CO fields to determine if a particular GOSAT sounding is likely to be fire affected. In addition to emissions from CO sources and their atmospheric transport, the CAMS CO fields incorporate carbon monoxide total column measurements from the IASI and MOPITT instruments (Inness et al., 2015). We sampled the CO fields at the time/location of each GOSAT sounding, and based on the CO distribution and data from the GOSAT Cloud and Aerosol Imager (CAI), values in excess of 0.003 kg/m$^2$ were determined as being likely affected by the local fire emissions. Conversely, if the CO value was less than 0.00075 kg/m$^2$ then the sounding was classed as "clear" (i.e. unaffected by local fire emissions). GOSAT soundings corresponding to locations and times having CO values between these thresholds were not able to be confidently classed as either "fire affected" or "clear". Out of 3946 GOSAT soundings over Indonesia during September/October 2015, the CAMS CO identifies 341 (8.6%) of these as being affected by fire and 1272 (32.3%) as clear (i.e. unaffected by fire), and with the remainder lying between these thresholds.

Figure 4 shows a GOSAT CAI false-colour image covering much of Kalimantan on 21st October 2015, a time when a massive pall of smoke enveloped Central Kalimantan and parts of the surrounding regions. The active fire detections for this day made from MODIS are also shown (small purple circles), along with the numbered locations of the individual GOSAT TANSO-FTS soundings (red circles). All GOSAT soundings made co-incident with this CAI image were in locations where the simultaneous CAMS CO field indicated the corresponding TANSO-FTS measurement was 'fire affected'.

## 5  Observations of Enhanced Methane Concentrations

Once we had identified a set of GOSAT soundings that were able to be clearly classed as "fire affected" or "clear", it became possible to examine the $XCH_4$, $XCO_2$ and $XCH_4/XCO_2$ distributions in order to determine the changes in the column amount and trace gas ratio characteristics related to the extreme levels of fire activity. Figure 5 shows histograms of the $XCH_4/XCO_2$





ratio, as well as the individual $XCH_4$ and $XCO_2$ components, for all the "clear" (blue) and "fire affected" (red) soundings, as well as for the entire dataset (green).

As Table 1 shows, for the $XCH_4/XCO_2$ ratio, the mean ratio calculated from all the data is 4.54 ppb/ppm, with a standard deviation of 0.033 ppb/ppm. The histograms for the clear and fire-affected data show two clearly separated distributions, with means of 4.52 and 4.59 ppb/ppm respectively. When examining just the $XCO_2$ distributions, there appears to be less of a distinct separation, with means of 399.9 and 401.1 ppm respectively for the clear and fire-affected cases. This corresponds to a $XCO_2$ increase of 0.3% percent over the background $XCO_2$ concentrations, whereas the $XCH_4$ distribution for the fire-affected scenes shows a much larger mean enhancement of 1.9% percent over the background (1840.1 ppb vs 1805.5 ppb).

In order to examine the spatial distribution of the atmospheric GHGs and $XCH_4/XCO_2$ ratio enhancements, Figure 6 shows (top to bottom) maps of the GOSAT-retrieved $XCH_4$, $XCO_2$, $XCH_4/XCO_2$ ratio, along with the CAMS total column CO and IASI total column CO for all TANSO-FTS sounding locations (left), "clear" locations (centre) and "fire-affected" locations (right). These data show that the spatial extent of the enhancements in $XCH_4$, $XCO_2$ and in the resulting $XCH_4/XCO_2$ ratio, as well as in the CAMS and IASI CO, are related to the enhanced fire activity seen over parts of Sumatra and Kalimantan (shown in Fig. 2), whose emissions are being transported primarily westwards over the ocean (last column of Fig. 6).

This finding confirms that the anomalously large amount of fire activity seen occurring in September and October 2015 during the El Niño (Fig. 1) and which included fires in the extensive peatlands of Central Kalimantan and south Sumatra (Fig. 2) resulted in a significant increase in atmospheric column amounts of $CH_4$ and $CO_2$ downwind of the fires. These enhancements are observable from GOSAT satellite observations, and in the following section we examine the $CH_4/CO_2$ emission ratio (ER) of this smoke to better understand the combustion characteristics.

## 6 Determination of $CH_4/CO_2$ Emission Ratios

As discussed in Section 2, the capability to determine large-scale regional emission ratios during intense fire-activity is important as it allows information to be gained not only on the emissions of these gases themselves but also potentially on the relative dominance of flaming vs. smouldering combustion. Our previous work Ross et al. (2013) demonstrated for the first time an ability to determine $CH_4/CO_2$ fire emission ratios from satellite data, in that case using GOSAT to study ERs of boreal forest (Canada/Russia), tropical forest (Brazil) and savannah (Southern Africa) fires. The satellite-derived emission ratios obtained appeared to be in good agreement with those derived during ground and aircraft sampling studies in the same biomes, albeit these in-situ data themselves show relatively large variations. Such variability is likely a function of differences in fuel type, fuel moisture and fire behaviour that occurred between different measurement campaigns, fire locations and time of year or day (Van Leeuwen and Van Der Werf, 2011). Here we apply the technique of Ross et al. (2013) to our current GOSAT Proxy retrievals of $XCO_2$ and $XCH_4$ made during the September-October 2015 Indonesian fires, in order to determine the emission ratios characterising the very large scale plumes seen during this anomalously large climate-related fire event.

As a first step in this process, it is useful to calculate the excess (or "$\Delta$") $XCH_4$ and $XCO_2$ values prior to any subsequent processing, since for example the fire emissions can be superimposed into a "background" atmosphere that itself contains



spatially and/or temporally varying amounts of $XCH_4$ and $XCO_2$. Calculating such excess amounts removes the impact of potentially varying background concentrations. However, since we utilise the $XCH_4$ and $XCO_2$ components of the GOSAT Proxy $XCH_4$ retrieval, which themselves do no account explicitly for aerosol scattering (but instead relying on these effects to ratio out when computing the final Proxy $XCH_4$ values; see Section 3) this does provide some potential for error to be

introduced in any subsequently calculated $CH_4/CO_2$ emissions ratio. Such errors are related to the fact that the degree of scattering may be different between the "fire affected" (i.e. smoke laden) and matching background (i.e "clear") TANSO-FTS soundings from which the excess amounts are calculated. We analysed the magnitude of this effect using a simple model, included as Appendix A, and the results indicate that it is possible to underestimate the $CH_4/CO_2$ emission ratio by around ∼10% if appropriate care is not taken during selection of the 'clear' soundings whose column amounts are to be subtracted

from those of the "fire affected" soundings in order to calculate the excess column amounts. In a region such as Indonesia during El Niño, where large-scale fire activity is clearly greatly affecting the aerosol composition of the local atmosphere, this aspect becomes even more challenging. To deal with this, we only used fire affected TANSO-FTS soundings made over land, so as to minimise the effect of mixing/dilution as smoke-laden air was transported longer distances over the ocean. For each fire-affected sounding a matching background measurement was selected from the group of "clear" soundings located

over the same island and as close as possible to the fire-affected measurement (e.g. the background for the Sumatra soundings were selected from clear soundings between 90°E-108°E and 5°N-10°S) in order to minimise impacts stemming from use of non-uniform background measurements as detailed in Yokelson et al. (2013). Out of 131 fire-affected soundings, a suitable background sounding was identified for 105 (80%) of the soundings. Each background $XCH_4$ and $XCO_2$ value was then subtracted from the concentration derived from its corresponding fire-affected sounding in order to produce the $\Delta XCH_4$ and

$\Delta XCO_2$ values, from which the emission ratios could then be calculated.

Figure 7 shows the $\Delta XCH_4$ values plotted against the simultaneously derived $\Delta XCO_2$ values for all of the fire-affected soundings measured over Indonesia for the September-October 2015 period. The $CH_4/CO_2$ emission ratio derived from the linear best fit to these data is 6.2 ppb/ppm (correlation coefficient of 0.937). Whilst this calculated $CH_4/CO_2$ emission ratio is significantly above that of the ambient background (∼4.52 ppb/ppm), the many fire plumes sampled by GOSAT soundings

across the September-October 2015 period mean that potential variations in the emissions ratios over time (and space) can also be explored. Figure 8 once again shows the daily CAMS FRP density data, but this time for September-October 2015 only and as a daily average for the entire Indonesian region as well as for Sumatra and Kalimantan individually. There is very significant variability seen in the fire activity across these two months, and we identify several distinct time periods to examine in more detail for both Sumatra and Kalimantan. The period 9th - 15th September over Sumatra is characterised

by a steady increase in FRP density, peaking on 12th September at over 200 W/m$^2$ before decreasing again and reducing to below 50 W/m$^2$ by 15th September and we take this as Period 1 for Sumatra. By contrast, over Kalimantan at around the same time (specifically between the 8th and 17th September) there is a peak in FRP on 8th September of nearly 300 W/m$^2$, followed by a lull around the middle period before a second increase to almost 150 W/m$^2$ on 13th/14th September. We take this as Period 1 for Kalimantan. In contrast to the differing behaviours during Period 1, both Sumatra and Kalimantan exhibit

somewhat more similar trends in fire activity during Period 2, starting with a high peak (over 300 W/m$^2$) on 19th and 14th





October for Sumatra and Kalimantan respectively, however while over Kalimantan the fire activity then immediately reduces to a lower level (around 100 W/m$^2$), over Sumatra high FRP density values in excess of 300 W/m$^2$ are maintained over several days before slowly decreasing. This suggests a significantly larger fire event over Sumatra than over Kalimantan at this time, a finding consistent with the CAMS total column CO fields (e.g. as seen in Fig. 6). Although Fig. 8 suggests that an additional

period centred around 22nd September should be of interest, there are insufficient GOSAT soundings during this time from which to determine an ER, demonstrating that the somewhat limited GOSAT sampling strategy can lead to a sparseness of observations in certain situations.

Figure 9 shows the $\Delta$XCH$_4$ vs $\Delta$XCO$_2$ measurements recorded over Sumatra, for the entire 2-month period (Sept - Oct) (top), and for Period 1 (middle) and Period 2 (bottom) only. Over the two months, a total of 66 fire-affected TANSO-FTS

measurements are identified that have a suitable matching background available from which to calculate $\Delta$XCH$_4$ and $\Delta$XCO$_2$. The linear best fit to these data give a CH$_4$/CO$_2$ emission ratio of 6.64 ppb/ppm (R = 0.893) for these Sumatran fires. When examining the Periods 1 and 2 only, which Fig. 8 shows corresponds to times of increased fire activity over the Island, higher emission ratios of 8.1 and 8.8 ppb/ppm are derived (R = 0.91 and 0.92 respectively). These higher CH$_4$/CO$_2$ emission ratios are consistent with the region being characterised by a larger proportion of smouldering combustion, most likely of peatland

given the preponderance of that landcover in the fire-affected area (Fig. 2), resulting in enhanced CH$_4$ concentrations as already observed in Section 5.

Similar to Figure 9, Figure 10 shows the $\Delta$XCH$_4$ and $\Delta$XCO$_2$ retrievals for Kalimantan, plotted on a scatterplot from which the CH$_4$/CO$_2$ emission ratio can be derived. Over the two months of September and October 2015, Fig. 8 shows that Kalimantan appears characterised by typically lower amounts of fire activity than Sumatra, interspersed with relatively

short but intense episodes such as those on 8th September and 14th October. The CH$_4$/CO$_2$ emission ratio calculated for the Kalimantan data across the entire two month period is found to be 6.2 ppb/ppm, calculated from 39 separate $\Delta$XCH$_4$ and $\Delta$XCO$_2$ observations (correlation coefficient of 0.974). However, when examining Period 1 only (8th-17th September), although derived from only 9 data points (R=0.92) an extremely high emission ratio is found (13.6 ppb/ppm, R = 0.92). By contrast, during Period 2 (14th-25th October) the ER is found once again to be lower, at 6.2 ppb/ppm (R = 0.97). This lower

value may be affected by the fact that throughout Period 2 extensive smoke aerosol covers much of Kalimantan (as seen in Fig. 4) and the selection of 'clear' TANSO-FTS that appropriately represent the clean background of the fire-affected measurements is significantly more difficult. This is further compounded by the fact that the wind vectors (not shown) for this period indicate that the background air is likely to be coming from further south, potentially having a different CH$_4$ and CO$_2$ concentration.

## 6.1   Ground Based Emission Ratios from El Niño Enhanced Peat and Vegetation Fires

In addition to space-based observations described above, during October 2015 at the height of the fire activity on Kalimantan (Fig. 8) a short field campaign was conducted to derive CH$_4$ to CO$_2$ emission ratios for comparison to the GOSAT-derived values. During this campaign, smoke was sampled and emission ratios derived for individual fire plumes stemming from the El Niño enhanced landscape fires. Trace gas mixing ratio measures of CO$_2$ and CH$_4$ were made at 1 Hz frequency in plumes from fires at four different locations within ∼30 km of Palankarya (2.21° S, 113.92° E), the capital of Central Kalimantan





and one of the most fire affected regions during the 2015 El Niño related drought (Drake, 2015). We use a ground-based, more portable version of the cavity enhanced laser absorption spectrometer described in O'Shea et al. (2013). The precision (Allan variance, 1 sigma @ 1 hz) of the mixing ratios derived via the laser spectroscopy was 1.71 ppb for $CH_4$, and 2.63 ppm for $CO_2$, with a total absolute uncertainty of around 1% of the measured concentrations. Fires at the four different locations

were sampled between 12th and 16th October 2015, with each site located on peat but with plumes encompassing both "pure" peat burning and also times when both peat and some overlying vegetation was being consumed. The $CH_4/CO_2$ emission ratios determined from these close-to-source measurements varied between 7.53 and 19.7 ppb/ppm (mean $\pm$ sd = 12.9 $\pm$ 3.9 ppb/ppm), a range relatively consistent with that determined from the GOSAT-derived space-based observations (6.18 to 13.6 ppb/ppm). However, the majority of the ground-based emission ratios were derived from locations dominated by almost pure

peat burning sampled close to source, whereas the space-based observations from GOSAT are derived from measurements of the smoke filling a 10.5 km diameter TANSO-FTS footprint and thus representative of much larger areas of combustion, very likely comprising a mix of peat and vegetation burning in the majority of cases.

Despite this potential for the GOSAT-derived $CH_4$ to $CO_2$ ER to be somewhat less characteristic of "pure" peat burning than are some of the "close to source" measurements, and the potential for the measurements to be influenced by cleaner

air (such as that transported from the south), it is still expected that the emissions over Indonesia will be largely dominated by smouldering combustion, resulting in a typically higher $CH_4/CO_2$ ER than that observed from flaming combustion as is generally characteristic of most African Savannah burning (Wooster et al., 2011). To confirm that this is the case, the same GOAST-based analysis performed here for the 2015 Indonesian fires was also performed for southern Africa (defined as 0°N-40°S, 30°W-60°E) and the Amazon (defined as 0°N-40°S, 30°W-75°W), both of which underwent significant fire activity

during this same time period. The calculated $CH_4/CO_2$ emission ratio for southern Africa was found to be 4.35 ppb/ppm (see Appendix B, Fig. 12), consistent with observations of flaming-dominated combustion in Savannah regions. Wooster et al. (2011), using a ground-based open path FTIR system, reported $CH_4$ to $CO_2$ ERs for different phases of southern African savannah burns conducted on 7 ha plots in Kruger National Park, South Africa. Backfires (spreading against the wind) typically produced emissions with very complete combustion characteristics, with $CH_4$ to $CO_2$ ERs of 1.9 - 2.2 ppb/ppm, apart from in

one case where a value of 6.0 ppp/ppm was recorded. Residual areas of smouldering combustion present after the fire front had passed were recorded as having $CH_4$ to $CO_2$ ERs of 3.1 - 14.1 ppb/ppm, although it was possible that the lowest ERs reported were significantly influenced by remaining pockets of flaming activity. The headfire emissions which combine the smoke from the most intense flaming part of the burn with those from the "smouldering zone" immediately behind were found to have $CH_4$ to $CO_2$ ERs of 2.4 - 5.4 ppb/ppm. The overall "fire averaged" $CH_4$ to $CO_2$ emission ratio, calculated from the ERs of the

individual phases and using airborne measures of FRP to estimate the amount the fuel consumption in each for the purposes of the weighting calculation, was 4.3 $\pm$ 1.7 ppb/ppm, very close to the 4.35 ppb/ppm derived from GOSAT's observations of large-scale southern African savannah plumes. This provides further evidence for the representivity of our GOSAT approach, which is currently the only method able to assess the ERs of the largest plumes emanating from landscape fires, albeit only at the relatively sparse sampling locations targeted by GOSAT. The $CH_4/CO_2$ emission ratio for the Amazon is, as perhaps

expected, somewhat in-between that of African savannah and Indonesian peatlands/forests, being 5.1 ppb/ppm (see Fig. 13).





Guild et al. (2004) and references therein report the presence of significant smouldering combustion in Amazon fires occurring in forested regions, much more than typically seen in African savannah and primarily stemming from the coarse woody fuels that represent a significant component of the fuel in this biome. However, smouldering in the peat-dominated fuels of the Indonesian fires would still be expected to be more prevalent (Stockwell et al., 2014), and so the $CH_4$ to $CO_2$ ER would be expected to be higher there, as we have indeed found.

## 7 Summary and Outlook

The objective of this study was to utilise $XCH_4$ and $XCO_2$ observations made by the GOSAT satellite when passing over Indonesia to probe the composition of large-scale plumes from the 2015 Indonesian fires for the first time, these extreme fires being driven by the ongoing strong El Niño that is the largest seen since 1997-98. We wished to both identify the atmospheric greenhouse gas impacts of the very significant increase in fire activity and use any such measurements to determine the biomass burning emission ratios of these two important GHGs using the technique we pioneered in Ross et al. (2013). This would enable the characterisation of certain aspects of the chemical make-up of these large-scale El Niño driven fires for the first time, which in 1997-98 were responsible for the largest release of fire emitted GHGs seen worldwide, and indeed which are believed to be of a magnitude not seen since that period anywhere on Earth (van der Werf et al., 2010).

Our analysis of GOSAT data confirms a significant enhancement of both $XCH_4$ and $XCO_2$ in the fire-affected GOSAT soundings, with the greatest change seen in the $XCH_4$ mixing ratios where we see an average value of 1840.1 ppb compared to an average value in the 'clear' (non-fire affected) cases of 1805.5 ppb. For these fire-affected soundings, the $CH_4/CO_2$ emission ratio was estimated from the gradient of the linear best fit to the excess $XCH_4$ and $XCO_2$ values. We find an overall ER for the entire Indonesian fire-affected region during the September-October 2015 fire peak of 6.2 ppb/ppm, with Sumatra showing slightly higher mean ERs (6.6 ppb/ppm) than Kalimantan (6.2 ppb/ppm). When examining shorter periods of time to focus on specific fire episodes on each island, we find ERs as low as 6.1 and as high as 13.6 ppb/ppm. This range is consistent with that seen in field-sampled GHG data taken in October 2015 on Kalimantan, close to the fire sources, but we believe the large-scale sampling provided by the GOSAT TANSO-FTS 10.5 km diameter footprints enables sampling of a much more representative amount of smoke than does the relatively limited, small-scale sampling possible on the ground. We therefore believe that our GOSAT-derived emission ratios are well suited for use in studies attempting to understand the impact of these extreme El Niño driven fires on the larger-scale, regional atmosphere.

Our GOSAT-derived emission ratios for Indonesia indicate plumes that appear more dominated by the products of smouldering combustion than the plumes sampled by GOSAT in southern Africa and in the Amazon during the same period, consistent with prior expectation and previous ground-based and airborne sampling campaigns that suggest less smouldering dominated combustion in these latter biomes (especially in the savannah case). GOSAT's capability to determine not only the enhancement in greenhouse gas abundance stemming from such large fire events, but also to provide the data necessary to calculate the GHG emissions ratios and help identify the relative balance of smouldering and flaming activity ongoing in very large regions is an extremely valuable aid to understanding the composition of the plumes and their impact on regional atmospheric





composition and climate. Some challenges remain, mainly relating to obtaining an accurate representation of the background "non-fire affected" $XCH_4$ and $XCO_2$ amounts (See Appendix A). However, the technique that we present here and in Ross et al. (2013) should be easily applicable to future satellite missions focused on atmospheric composition, and certain of these will have increased spatial and temporal resolutions that will greatly aid in obtaining the most suitable background observa-

tions. One such mission, Sentinel-5 Precursor, is planned for launch in 2016 and is capable of measuring both $CH_4$ and CO at a high spatial resolution, providing an ability that GOSAT currently lacks. We believe therefore that this work will prove valuable in eventually facilitating the routine determination of regional biomass burning ERs from space, and their spatio-temporal variations whose importance is described in e.g. Van Leeuwen and Van Der Werf (2011). Such a capability might ultimately allow the characterisation of such burning events under different climatic and biome conditions.

**Appendix A**

As discussed in Section 6, there exists the potential to introduce errors into the GOSAT-derived emission ratios if the fire-affected sounding and the background sounding each contain sufficiently different aerosol scattering characteristics. As the fire-affected sounding will by definition usually contain a non-trivial amount of smoke aerosols, whilst the background sounding is in theory supposed to be smoke-free, some quantification of this affect is needed. In this section we derive and use a simple

mathematical representation to determine the magnitude of such effects.

Let the observed excess concentration be the difference between the observed fire and background concentrations:

$$\Delta XCH_4 = XCH_4^{fire} - XCH_4^{bgd} \tag{A1}$$

Both soundings will have an error due to scattering associated with them which typically lengthens the light-path and hence reduces the inferred gas mixing ratio. This error factor, here termed A, will be different for the fire and background cases.

Therefore:

$$\Delta XCH_4 = A_{fire}XCH_4^{fire} - A_{bgd}XCH_4^{bgd} \tag{A2}$$

Similarly for $XCO_2$ we have:

$$\Delta XCO_2 = A_{fire}XCO_2^{fire} - A_{bgd}XCO_2^{bgd} \tag{A3}$$

The ratio of the excess concentrations due to the fire emissions, from which we calculate the $CH_4$ to $CO_2$ emission ratio, is

then given by:

$$\frac{\Delta XCH_4}{\Delta XCO_2} = \frac{A_{fire}XCH_4^{fire} - A_{bgd}XCH_4^{bgd}}{A_{fire}XCO_2^{fire} - A_{bgd}XCO_2^{bgd}} \tag{A4}$$

Now we set the observed CH4 concentration in the fire sounding to the background concentration plus the true excess concentration related to the fire:

$$XCH_4^{fire} = XCH_4^{bgd} + \Delta XCH_4^{true} \tag{A5}$$





Doing the same for $XCO_2$ now gives:

$$\frac{\Delta XCH_4}{\Delta XCO_2} = \frac{A_{fire}(XCH_4^{bgd} + \Delta XCH_4^{true}) - A_{bgd}XCH_4^{bgd}}{A_{fire}(XCO_2^{bgd} + \Delta XCO_2^{true}) - A_{bgd}XCO_2^{bgd}} \tag{A6}$$

This can then be expanded to:

$$\frac{\Delta XCH_4}{\Delta XCO_2} = \frac{A_{fire}XCH_4^{bgd} + A_{fire}\Delta XCH_4^{true} - A_{bgd}XCH_4^{bgd}}{A_{fire}XCO_2^{bgd} + A_{fire}\Delta XCO_2^{true} - A_{bgd}XCO_2^{bgd}} \tag{A7}$$

and then rearranged to:

$$\frac{\Delta XCH_4}{\Delta XCO_2} = \frac{(A_{fire} - A_{bgd})XCH_4^{bgd} + A_{fire}\Delta XCH_4^{true}}{(A_{fire} - A_{bgd})XCO_2^{bgd} + A_{fire}\Delta XCO_2^{true}} \tag{A8}$$

Now let the ratio of the two error terms be

$$A_{ratio} = A_{fire}/A_{bgd} \tag{A9}$$

which then rearranged gives:

$$A_{fire} = A_{ratio}A_{bgd} \tag{A10}$$

Therefore

$$A_{fire} - A_{bgd} = A_{ratio}A_{bgd} - A_{bgd} = A_{bgd}(A_{ratio} - 1) \tag{A11}$$

Substituting this in now gives:

$$\frac{\Delta XCH_4}{\Delta XCO_2} = \frac{(A_{bgd}(A_{ratio} - 1))XCH_4^{bgd} + A_{ratio}A_{bgd}\Delta XCH_4^{true}}{(A_{bgd}(A_{ratio} - 1))XCO_2^{bgd} + A_{ratio}A_{bgd}\Delta XCO_2^{true}} \tag{A12}$$

The $A_{bgd}$ terms cancel, giving:

$$\frac{\Delta XCH_4}{\Delta XCO_2} = \frac{(A_{ratio} - 1)XCH_4^{bgd} + A_{ratio}\Delta XCH_4^{true}}{(A_{ratio} - 1)XCO_2^{bgd} + A_{ratio}\Delta XCO_2^{true}} \tag{A13}$$

This equation therefore relates the observed excess concentrations ($\Delta XCH_4$ and $\Delta XCO_2$) calculated from the difference in GOSAT's "fire-affected" and "background" soundings to the true background concentrations ($XCH_4^{bgd}$ and $XCO_2^{bgd}$), the true excess concentrations ($\Delta XCH_4^{true}$ and $\Delta XCO_2^{true}$) and the ratio between the error terms, $A_{ratio}$. Furthermore, we can use this simple relationship to explore the likely error in the calculated emission ratio for a given value of $A_{ratio}$.

Figure 11 shows the implementation of Equation A13 for various scenarios. The background $XCH_4$ and $XCO_2$ concentrations are fixed at 1850 ppb and 400 ppm respectively, representing the normal "fire-free" atmosphere. The true methane enhancement ($\Delta XCH_4^{true}$) is varied between 0-50 ppb in 5 ppb increments and the true emission ratio between $CH_4$ and $CO_2$ is varied between 0.003 and 0.012. The different panels then show the behaviour for various ranges of $A_{ratio}$. The top-left panel





has $A_{ratio}$ set at a constant value of 1 (i.e. the error in the background is exactly the same as the error in the fire cases), which is the ideal situation, and the true emission ratios are reproduced exactly. The top-right panel allows $A_{ratio}$ to vary between 0.999 and 1.0. The effect of this is a slight "spreading" of the lines and the difference between the true and observed emission ratio is minimal. The bottom-left panel increased the range of $A_{ratio}$ to 0.995 to 1.0 which causes the observed emission ratios to

differ more from the truth, with a true emission ratio of 0.008 only appearing as an observed ratio of 0.00755, an error of 5.6%. Finally, in the bottom-right panel, the value of $A_{ratio}$ is allowed to vary between 0.99 and 1.0. This relatively large variation decreases the observed emission ratio further, with a true emission ratio of 0.008 appearing as an observed ratio of 0.00686 for example.

Whilst there are many unknowns that impact the value of $A_{ratio}$, and so it is not possible to know its exact value for a

particular pair of GOSAT "fire affected" and "background" observations used to derive an emissions ratio, it is possible to determine its expected range. By comparing the scatter of the fire affected and background $XCH_4$ values to the Proxy $XCH_4$ data (which is much less affected by aerosol and in this case is used as the "truth") it is possible to estimate the likely range of values of $A_{ratio}$. The standard deviation of the ratio between the $XCH_4$/Proxy $XCH_4$ for the background and fire cases is found to be 0.00494, suggesting that values of $A_{ratio}$ are likely in the 0.995 to 1.0 range (i.e. up to a 0.5% reduction in $A$). This

means that whilst we are likely to tend to underestimate the true $CH_4$ to $CO_2$ emissions ratios with GOSAT, for the majority of cases ($CH_4$ to $CO_2$ ERs in the range 0.005-0.008) the effect can be considered small, with typical biases of 0.4-5.6%. Even in extreme cases with high ERS (e.g. 0.012), we expect an error of less than 15%.

## Appendix B

This section contains $\Delta XCH_4$ vs $\Delta XCO_2$ correlation plots for southern Africa (Fig. 12) and the Amazon (Fig. 13) as discussed

in the main text in Section 6.

*Acknowledgements.* R. J. Parker is funded via an ESA Living Planet Fellowship with additional funding from the UK National Centre for Earth Observation (NCEO) and the ESA Greenhouse Gas Climate Change Initiative (GHG-CCI). H. Boesch, D. Moore and M. Wooster are also supported by NERC NCEO. A, J. Webb is funded by a NERC PhD. Field measurements in Indonesia were part supported by NERC grant NE/J010502/1 [NERC SAMBBA]. Field measurements were also part supported by a DFID grant to CIFOR (Project No. 203034).

Bruce Main at King's College London and Agus Salim at CIFOR are thanked for their technical contributions to the field measurement campaign. D. Murdiyarso acknowledges the support provided by USAID and USFS. This work also relates to NERC Grant NE/N01555X/1.

We thank the Japanese Aerospace Exploration Agency, National Institute for Environmental Studies, and the Ministry of Environment for the GOSAT data and their continuous support as part of the Joint Research Agreement. We also thank CAMS for provision of the data from GFAS. We also acknowledge the use of MODIS Active Fire Detections extracted from MCD14ML distributed by NASA FIRMS

(available on-line [https://earthdata.nasa.gov/active-fire-data]). We thank EUMETSAT for the IASI CO Level 2 data. IASI is a joint mission of EUMETSAT and the Centre National d'Études Spatiales (CNES, France).

Finally, this research used the ALICE High Performance Computing Facility at the University of Leicester.





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





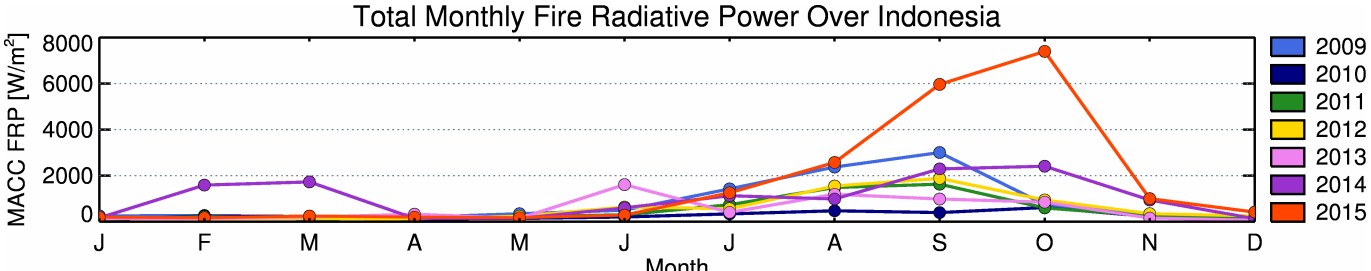

**Figure 1.** Time series of the monthly total Fire Radiative Power density (W/m$^2$) recorded over the Indonesian region (defined as 5°N-10°S, 90°E-150°E) between 2009 and 2015, calculated using data from the Copernicus Atmosphere Monitoring Services (CAMS') Global Fire Assimilation System (Kaiser et al., 2012). September and October 2015 are clearly anomalous compared to the previous years shown, highlighting the effect of this year's El Niño on the regions fire activity.

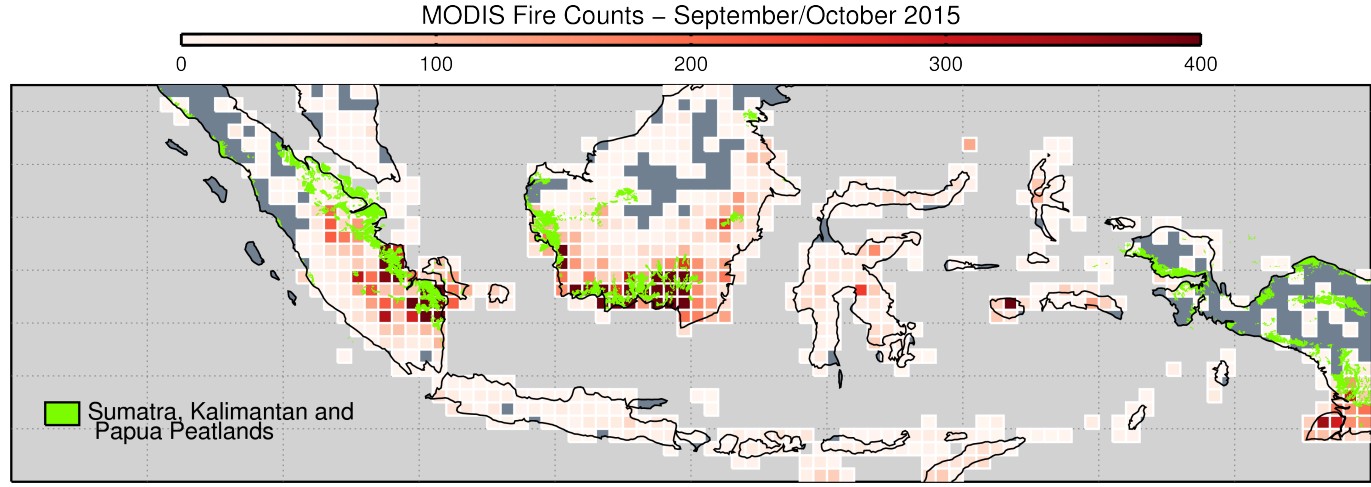

**Figure 2.** MODIS fire counts for September/October 2015 over the Indonesia, gridded into 0.5°x0.5° boxes. Also overlaid are the locations of known peatlands in Sumatra (left), Kalimantan (centre) and Papua (right).



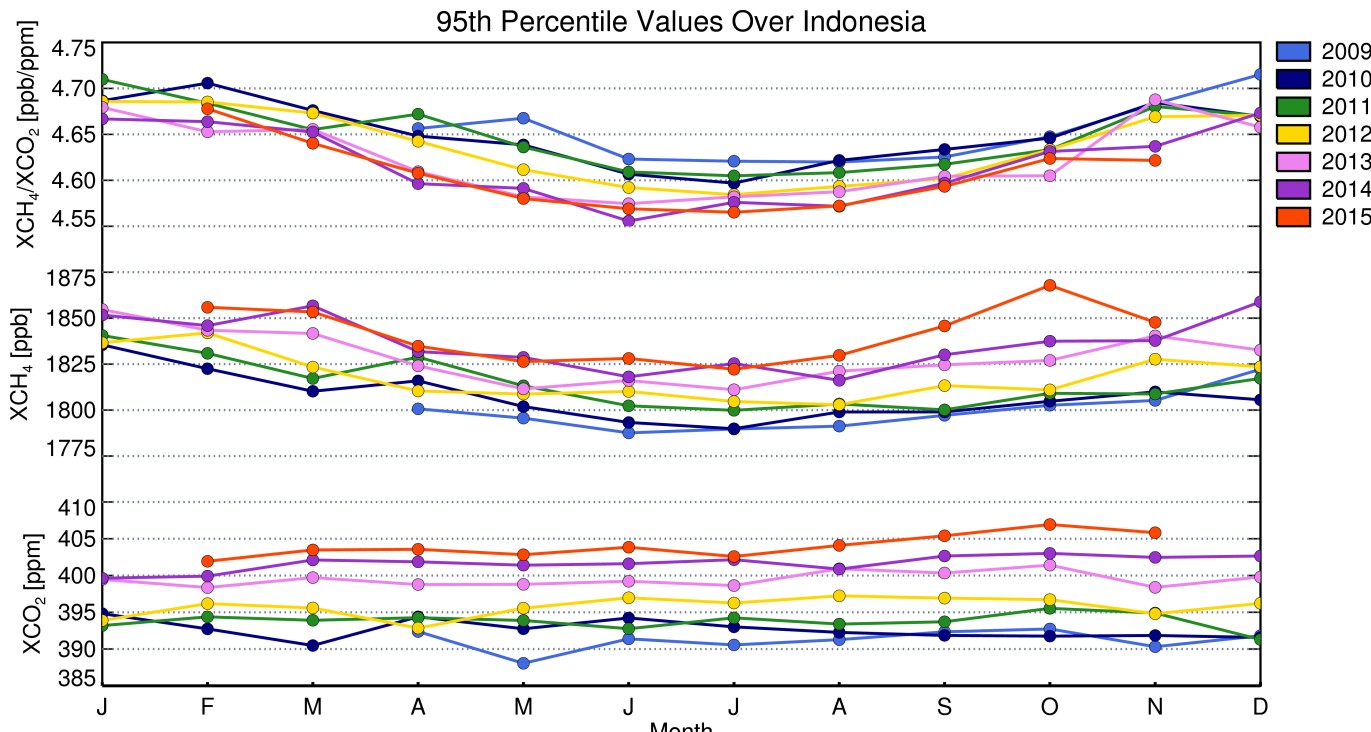

**Figure 3.** Timeseries showing the monthly 95th-percentile values over Indonesia for the GOSAT Proxy $XCH_4/XCO_2$ (top) as well as the individual $XCH_4$ (middle) and $XCO_2$ (bottom) components of the Proxy data for the entire GOSAT data record (2009-present).

**Table 1.** Table showing the mean and standard deviation over Indonesia in September/October 2015 of the $XCH_4/XCO_2$ Ratio (left), the retrieved $XCO_2$ (centre) and the retrieved $XCH_4$ (right) for all data (green), data determined to be unaffected by fire (blue) and data determined to be affected by fire (red).

| | $XCH_4/XCO_2$ (ppb/ppm) | | $XCO_2$ (ppm) | | $XCH_4$ (ppb) | |
|---|---|---|---|---|---|---|
| | Mean | Std. Dev | Mean | Std. Dev | Mean | Std. Dev |
| All | 4.54 | 0.033 | 399.9 | 4.88 | 1814.4 | 25.14 |
| Clear | 4.52 | 0.023 | 399.9 | 4.02 | 1805.5 | 18.43 |
| Fire-affected | 4.59 | 0.035 | 401.1 | 7.54 | 1840.1 | 37.83 |



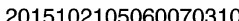

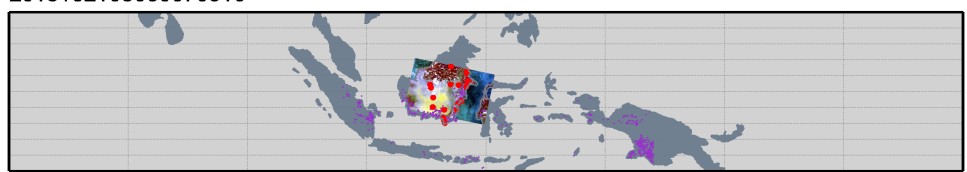

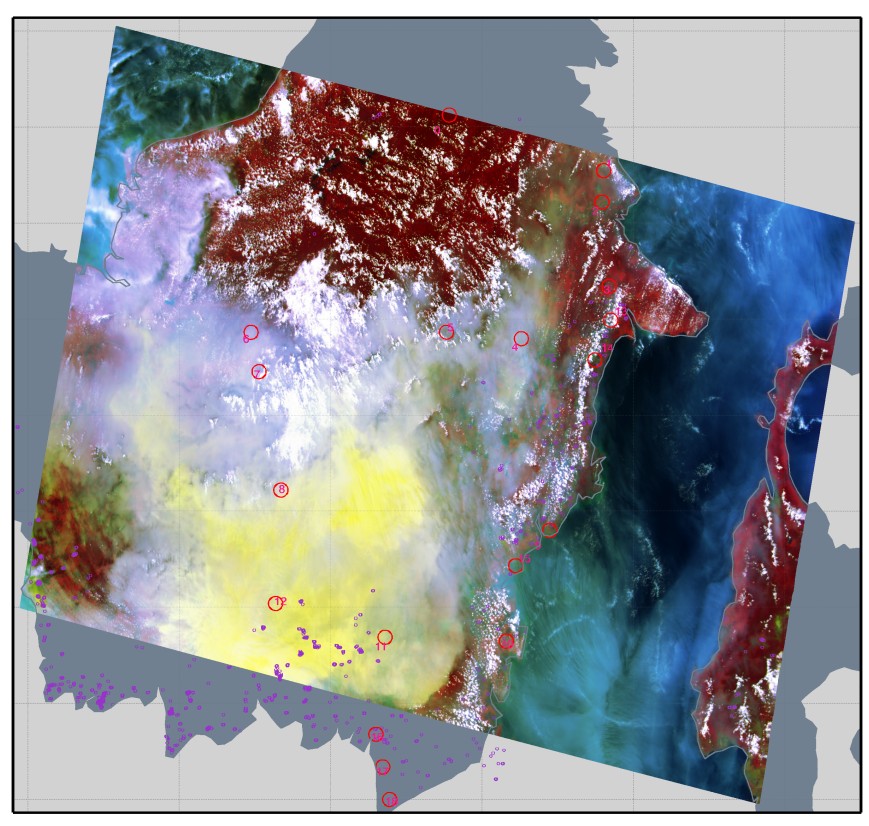

**Figure 4.** False-colour image (RGB = CAI Band 3, 2, 1) derived from data taken by the GOSAT Cloud and Aerosol Imager, collected when the GOSAT satellite passed over the island of Borneo on 21st October 2015 (around 1pm local time, 5am UTC), a period when extreme fires were burning across much of Central Kalimantan. GOSAT TANSO-FTS sounding locations are identified by the numbered large red circles, with the MODIS active fire detections identified by the small purple circles.



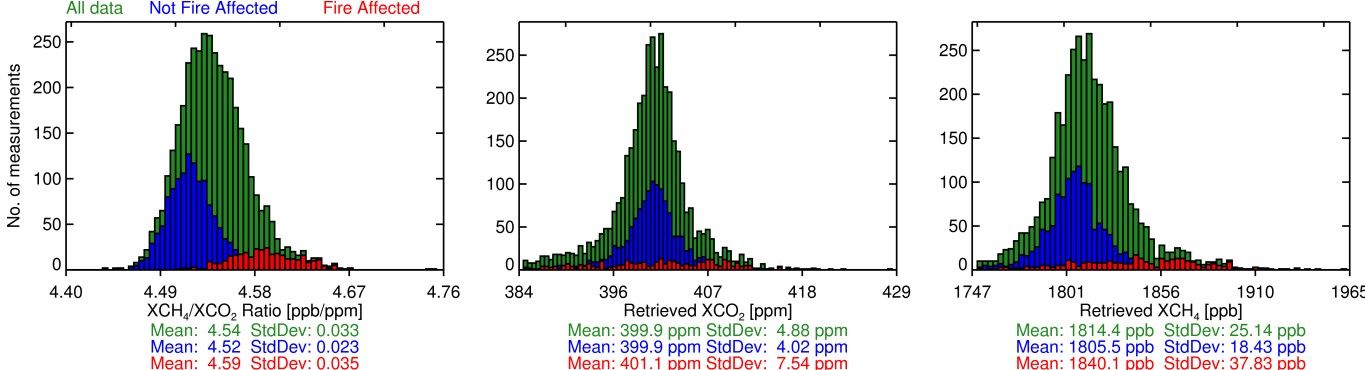

**Figure 5.** Histograms showing the distributions over Indonesia in September/October 2015 of the $XCH_4/XCO_2$ Ratio (left), the retrieved $XCO_2$ (centre) and the retrieved $XCH_4$ (right) for all data (green), data determined to be definitely fire-affected by fire emissions (red), and that classed as "clear" (blue). Also included are the corresponding mean and stdev values for each distribution.

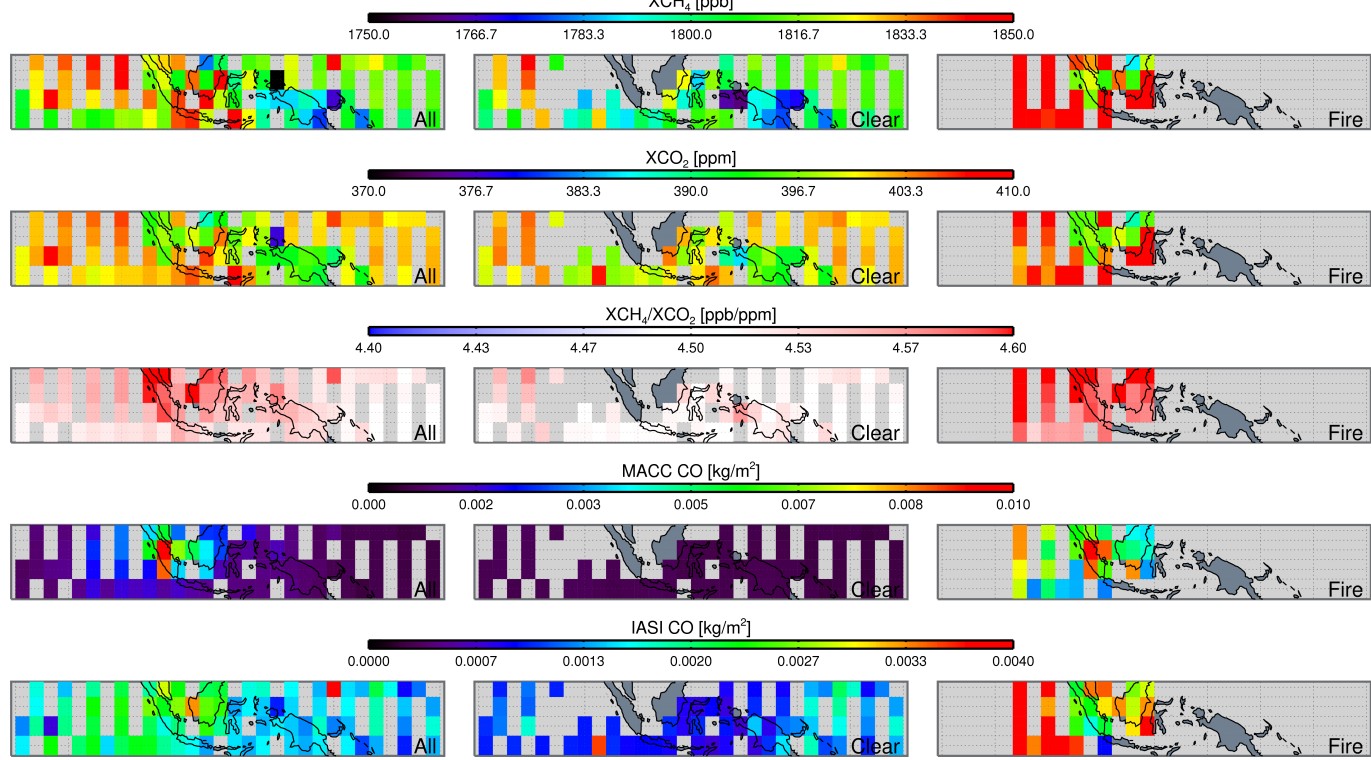

**Figure 6.** Indonesia trace gas distributions for September-October 2015 showing (top to bottom): the GOSAT-retrieved $XCH_4$, $XCO_2$, and $XCH_4/XCO_2$ ratio, along with the CAMS carbon monoxide (CO) total column and the measured IASI CO total column. The left column shows all data gridded at $2° \times 2°$ degrees , the central column shows only those points determined to be "clear" using the criteria of Section 4, and the right column shows the data determined to be fire affected based on the same criteria.



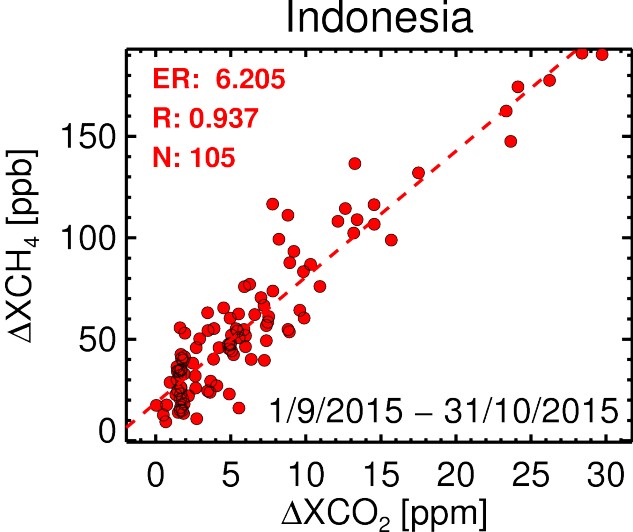

**Figure 7.** Scatterplot of GOSAT-derived $\Delta XCH_4$ vs $\Delta XCO_2$ values for large-scale fire plumes seen over Indonesia (of the type seen in Figure 4) from 1st September region from 1st September 2015 to 31st October 2015, calculated as the total column difference between the 'fire affected' and corresponding clear 'background' TANSO-FTS soundings. The $CH_4/CO_2$ emission ratio, ER (ppb/ppm), is calculated from the gradient of a linear best fit, shown as the dashed line. Also shown are the correlation coefficient, R and the number of soundings, N.

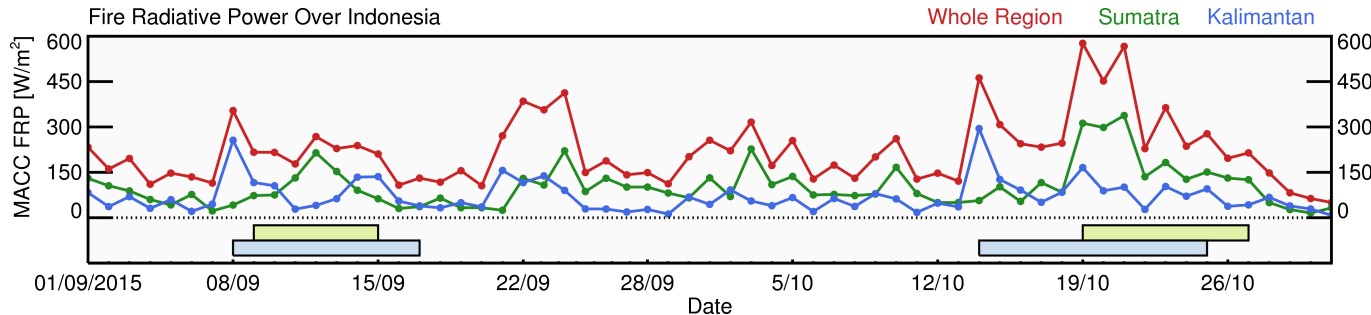

**Figure 8.** Daily Fire Radiative Power density (red line) taken from the Global Fire Assimilation System (GFAS) (Kaiser et al., 2012), operated as part of the Copernicus Atmosphere Service (CAMS). Data are shown from 1st September to 31st October 2015, for both the entire Indonesian landmass (red) and separately for the regions of Sumatra and Kalimantan. Two specific time periods are highlighted (referred to as Period 1 and Period 2), Period 1 covering 9th-15th September (Sumatra) and 8th-17th September (Kalimantan), and Period 2 covering 19th-27th October (Sumatra) and 14th-25th October (Kalimantan)

.





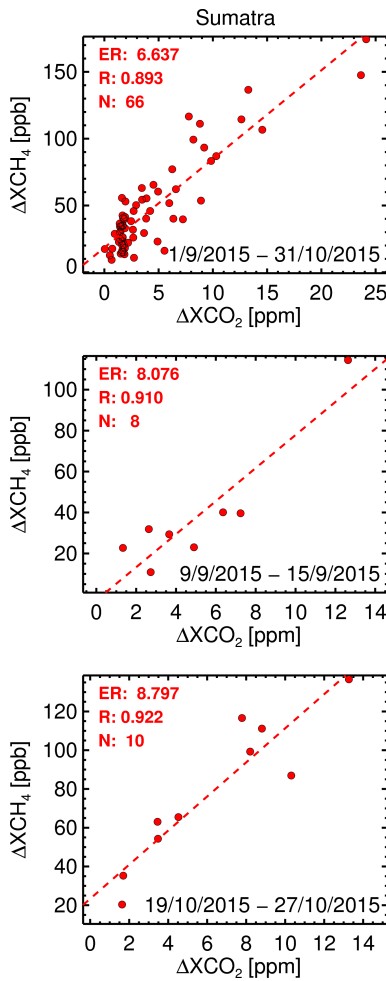

**Figure 9.** Scatterplots of $\Delta XCH_4$ vs $\Delta XCO_2$ derived for Sumatran large-scale fire plumes via analysis of TANSO-FTS data for the time periods detailed in Fig. 8: Sept-Oct 2015 (top), Period 1: 9th-15th September (middle), and Period 2: 19th-27th October (bottom). The $CH_4/CO_2$ emission ratio is calculated as the gradient of a linear fit to the data (dashed line). The correlation coefficient R and the number of soundings N are also shown.





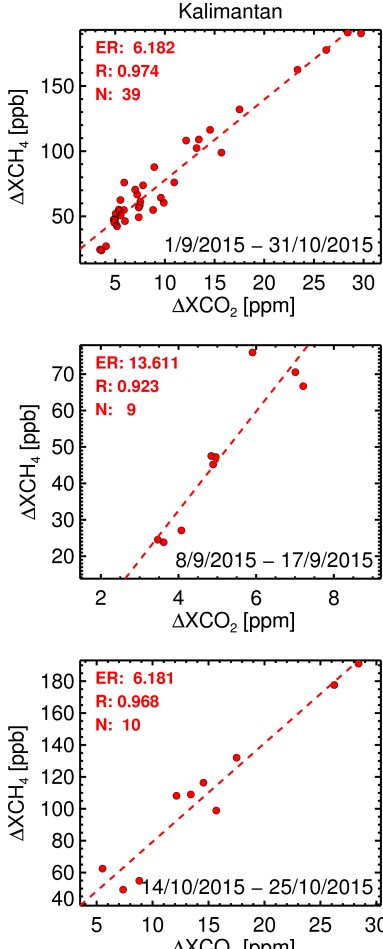

**Figure 10.** Scatterplots of $\Delta XCH_4$ vs $\Delta XCO_2$ derived for Kalimantan large-scale fire plumes via analysis of TANSO-FTS data for the time periods detailed in Fig. 8: Sept-Oct 2015 (top), Period 1: 8th-17th September (middle), and Period 2: 14th-25th October (bottom). The $CH_4/CO_2$ emission ratio is calculated as the gradient of a linear fit to the data (dashed line).The correlation coefficient R and the number of soundings N are also shown.





**Figure 11.** Implementation of Equation X with $\Delta XCH_4$ varied between 5-50 ppb for ERs ranging from 0.003 to 0.009 for different ranges of $A_{ratio}$. The true emission ratios and the emission ratios derived from the observed correlation are shown in each panel. The top left figure shows a fixed value of $A_{ratio} = 1$ while the remaining panels (clockwise) show the value of $A_{ratio}$ ranging from 0.999 to 1, 0.99 to 1 and 0.995 to 1.





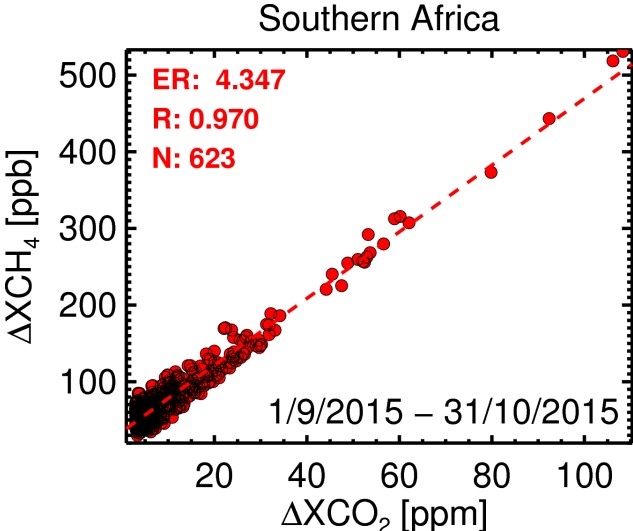

**Figure 12.** Scatterplot showing the $\Delta XCH_4$ vs $\Delta XCO_2$ values, calculated as the difference between the values in the fire-affected soundings to those in the background cases over the entire southern African region for 1st September 2015 to 31st October 2015. The $CH_4/CO_2$ emission ratio is calculated from the gradient of the linear best fit, which is shown along with the correlation coefficient R and the number of sounding pairs N. This GOSAT-derived ER is very similar to the 'fire averaged' $CH_4$ to $CO_2$ ER of $4.3 \pm 1.7$ ppb/ppm derived by Wooster et al. (2011) using open path FTIR spectroscopy measurements close to source on these type of savannah fire events. This line of best fit is also shown, along with the correlation coefficient, R and the number of sounding, N.





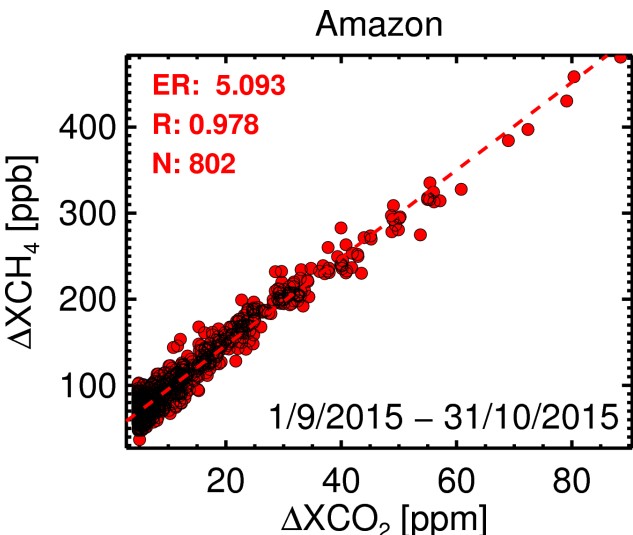

**Figure 13.** Scatterplot showing the $\Delta XCH_4$ vs $\Delta XCO_2$ values, calculated as the difference between the values in the fire-affected soundings to those in the background cases over the entire Amazonian region for 1st September 2015 to 31st October 2015. The $CH_4/CO_2$ emission ratio is calculated from the gradient of a linear fit to the data. This line of best fit is also shown, along with the correlation coefficient, R and the number of soundings, N.