# Peer review of "Atmospheric CH4 and CO2 enhancements and biomass burning emission ratios derived from satellite observations of the 2015 Indonesian fire plumes"

_Atmospheric Chemistry and Physics, 2016_

## Referee Comment (RC1) · Anonymous Referee #1 · 18 May 2016

Using the current 2015/16 El-Nino as a case study, the authors describe the use of GOSAT measurements of vegetation fire smoke plumes to characterise their chemical composition. Building on previous research, the authors demonstrate a technique to calculate the CH4/CO2 emission ratio which is indicative of the degree of smouldering to flaming combustion. The authors illustrate the potential of this approach and do so using observations acquired during the recent El-Nino event.

The manuscript is clearly written and concisely describes the approach for calculating the CH4/CO2 emission ratio and for assessing the uncertainty. The manuscript is

suitable for publication in ACP and its content will be of interest to the user community. It is recommended that it be published and included below are some minor comments.

Comments

The higher emission ratios, particularly those in Kalimantan, suggest a larger degree of smouldering to flaming combustion which may result from a greater number of peatland/forest fires. Figure 2 shows that grid cells which contain peatland often have higher fire counts associated with them. Some studies (e.g. Kaiser et al., 2012) have shown that peatland fires can have similar intensities during the day and night [which could partly explain the high fire count in these grid cells]. Do the authors have a view on the possible fraction of peatland/forest fires relative to agricultural burning ?

p1, line 14 : delete 'use'

p2, line 31 : El-Nino

p3, line 3 and 28 : delete the brackets around the year in references

p4, line 31 : delete the brackets around the year in references

P6, line 20 : Include the source of the peatland dataset (used in Fig. 2).

P9, line 2 : Is July used to determine the degree of enhancement (relative to October) as this is defined as the first month of fire activity (Figure 1)?

p10, line 29 and p15, line 8 : van der Werf

p11, line 3 : do not

p13, line 16 : GOSAT

p28 (Fig 7) : delete 'from 1st September region'

p31 (Fig 11) : replace 'Equation X' with 'Equation A13'.

P 31 (Fig 12) : Delete the last line (a largely duplicate sentence).

---

## Referee Comment (RC2) · Anonymous Referee #2 · 19 May 2016

**<General Comments>**

This paper demonstrated temporal and regional CH4 and CO2 measurement from space. This capability showed the detection of large-scale biomass-burning from space for the first time. It also suggested fuel type detection. I recommend publication after minor revision. I understand number of good in-situ data is limited, but discussion on rough estimation of horizontal and vertical distribution of the plume is helpful to understand the usefulness of large footprint (10km), point-based and, column averaged observations of GOSAT. Authors used CO data to identify the fire affected area. Fig-
ure 6 shows the comparison between XCH4/XCO2, MACC CO, and IASI CO. Further analysis or discussion on correlation between GOSAT-retrieved XCH4/XCO2 and CO will be useful.

<Specific Comments>

(1) Page 8, line 26 Description XCO2 retrieval method

My understanding is that the Proxy XCH4/XCO2 in this paper uses XCO2 from GOSAT instead of model XCO2. For CO2 retrieval do the authors use both 1.6 and 2.0 micron bands or only 1.6 micron band, which is closer to the CH4 band? The brief explanation is helpful.

(2) Page 23 line 16, Description on CAI will help reader's understanding.

CAI has a UV band at 380nm, which is shorter than MODIS blue band. CAI image helps distinguish aerosol absorption from cloud. From CAI image such as Figure 4, the spatial scale fire affected area can be estimated. Is there difference in CH4/CO2 between white and yellow (more absorption in UV)?

(3) Page 10, Line 10 and Page 27 Figure 6.

How do the authors retrieve XCO2 and XCH4 over dark ocean? Do the authors use glint data of TANSO-FTS?

(4) Page 15, Line 5

Authors mentioned high spatial resolution and imaging capability of Sentinel 5. Vertical profile information clarify the difference between in-situ CH4/CO2 and satellitemeasured column CH4/CO2. CH4/CO2 discussion using profile information from TANSO-FTS TIR band in the future might help.

<Technical Corrections>

(1) Page 5 line 31, fire radiative power (FRP).
Fire radiative power (FRP) is described already in Page 4, line 10.

(2) Page 8, line 5

Cryo-cooler restart is not "August 2015" (it stopped in August 2015) but "September 2015".

(3) Page 9, line 16, GOSAT Cloud and Aerosol Imager (CAI)

It already appeared as the Cloud and Aerosol Imager (TANSO-CAI) in Page 7 Line 29.

(4) Page 16, Equations (A6) and (A7)

Equations (A6) and (A7) are redundant.

(5) Page 25, Table 1

"Blue", "green" and "red" in Table 1 caption are not defined. Are these ones in Figure 8?

(6) Page 31, Figure 11, Upper left "Aratio = 1.0000...1" is confusing. Is it "Aratio=1.0"?

---

## Author Comment (AC1) · 22 Jun 2016

We would like to thank Referee 1 for taking the time to review our manuscript and appreciate the useful comments/corrections.

**Major comments:**

**Do the authors have a view on the possible fraction of peatland/forest fires relative to agricultural burning?**

Whilst this is difficult to quantify exactly, there has been some work in this area and

approximately 3/4 of the fuel consumption is estimated to have occurred as peatland burning. For further details, please see Huijnen et al., (2016) which includes some of our co-authors from this work. In the Supplementary Information of that paper the authors report that "From our total emissions estimate of 692 Tg CO2 produced in Sept-Oct 2015 over the region, 79

We have added a section in the text to discuss this as well as including the relevant reference.

"It is estimated that approximately $\frac{3}{4}$ of the fire activity over this time period was due to peatland burning (Huijnen et al., 2016)."

Added in reference to:

Huijnen, V. et al. Fire carbon emissions over maritime southeast Asia in 2015 largest since 1997. Sci. Rep. 6, 26886; doi: 10.1038/srep26886 (2016).

**Minor comments/typographical:**

**Various typographical corrections**

All typographical corrections have been fixed as recommended.

**Is July used to determine the degree of enhancement (relative to October) as this is defined as the first month of fire activity (Figure 1)?**

Yes. We'll clarify that in the text to better explain.

"In order to quantify the extreme nature of the October 2015 observations and to account for the annual growth rate, we define the magnitude of the enhancement as the October-July difference for each year, with July typically signifying the start of the fire season in this region".

---

## Author Comment (AC2) · 22 Jun 2016

We would like to thank Referee 2 for taking the time to review our manuscript and appreciate the useful comments/corrections.

**Major comments:**

**I understand number of good in-situ data is limited, but discussion on rough estimation of horizontal and vertical distribution of the plume is helpful to understand the usefulness of large footprint (10km), point-based and, column averaged observations of GOSAT.**

[Figure]

During the extreme fire activity observed, plumes ranged in scale from small, low, isolated plumes to huge plumes covering much of the landmass. We have attempted to capture the extent of this large scale behaviour by including Figure 4, showing the GOSAT Cloud and Aerosol Imager data over the region.

It should be noted that whilst the GOSAT data is a column quantity, the shortwave infrared measurements are most sensitive to the surface and lower atmosphere (unlike for example the thermal infrared IASI measurements which are mainly sensitive to the mid-troposphere). Also as the referee notes, the sampling pattern of GOSAT is not necessarily suited to making such point-source measurements and future satellites with imaging capabilities such as Sentinel-5 Precursor would be more suited to this. However, as this work focuses on attempting to quantify the large-scale behaviour of the entire region, we believe that, while challenging, we were successful in identifying GOSAT soundings dominated by the fire emissions.

**Further analysis or discussion on correlation between GOSAT-retrieved XCH4/XCO2 and CO will be useful.**

Whilst we acknowledge that being able to use co-located CO measurements along with the CH4 and CO2 would be extremely valuable and aid in calculating emission factors, there were several issues that prevented us from being confident in doing so.

Firstly, CO is not available from GOSAT. The best option would likely be to use the IASI CO product however, there are certain issues in doing so. Firstly, the IASI L2 CO data available from Eumetsat had an undocumented bug in the data product, with any retrieved integrated column CO values above 4.0e-3 kg/m2 being flagged as "invalid". Due to the huge extent of these Indonesian values, this upper limit is regularly exceeded and means that no quantitative comparison can be done against this IASI data (although it can still be used qualitatively as we have done). We have passed this information on to Eumetsat and believe that they are currently fixing this bug in their product. Secondly, however, IASI has very different vertical sensitivity to GOSAT which

means even if we were able to confidently co-locate soundings (with different overpass times), the comparison would be complicated to interpret. Whilst these issues can be overcome (e.g. through the use of assimilation into a chemistry transport model), we felt that this was potentially better suited to a future study. It should also be noted that GOSAT-2 (scheduled for launch in 2018) will be capable of measuring co-located CH4, CO2 and CO simultaneously.

**Minor Comments**

All typographical changes/recommendations have been included.

**For CO2 retrieval do the authors use both 1.6 and 2.0 micron bands or only 1.6 micron band, which is closer to the CH4 band?**

We only use the 1.6um CO2 band. This is described in more detail in our previous publications (e.g. Parker et al., 2015) and already stated in the text.

"this proxy method utilises the fact that the majority of the influence of atmospheric scattering on the retrieved XCH4 can be negated through the co-retrieval of the spectrally close 1.6 $\mu$m CO2 band"

**Description on CAI will help reader's understanding**

We have added the following section to the text to provide more details.

"The second instrument is the Cloud and Aerosol Imager (TANSO-CAI), which provides multispectral imagery at 0.5 km resolution with bands at 0.38 $\mu$m, 0.67 $\mu$m, 0.87 $\mu$m and 1.6 $\mu$m. This allows additional cloud/aerosol information about the region of interest within which the TANSO-FTS measurement footprints fall."

**Do the authors use glint data of TANSO-FTS?**

Yes, we use the sun-glint observations over the ocean. We will make this more explicit in the text.

[Figure]

"Whilst primarily performed over land, retrievals are also possible over the ocean when GOSAT measures in a sunglint geometry"

**Vertical profile information clarify the difference between in-situ CH4/CO2 and satellite measured column CH4/CO2. CH4/CO2 discussion using profile information from TANSO-FTS TIR band in the future might help.**

Please see this section in the text:

"However, the majority of the ground-based emission ratios were derived from locations dominated by almost pure peat burning sampled close to source, whereas the space-based observations from GOSAT are derived from measurements of the smoke filling a 10.5 km diameter TANSO-FTS footprint and thus representative of much larger areas of combustion, very likely comprising a mix of peat and vegetation burning in the majority of cases."

Regarding the TIR measurements from GOSAT, we don't believe that the data quality of these products is yet at a level to be useful due to calibration issues with the spectra. In the future, or for example from GOSAT-2, being able to separate out vertical profile information through combination of the SWIR and TIR bands would prove useful for this type of work.